# Why Do Artificially Generated Data Help Adversarial Robustness?

**Yue Xing**
Department of Statistics
Purdue University
xing49@purdue.edu

**Qifan Song**
Department of Statistics
Purdue University
qfsong@purdue.edu

**Guang Cheng**
Department of Statistics
University of California, Los Angeles
guangcheng@ucla.edu

## Abstract

In the adversarial training framework of Carmon et al. (2019); Gowal et al. (2021), people use generated/real unlabeled data with pseudolabels to improve adversarial robustness. We provide statistical insights to explain why the artificially generated data improve adversarial training. In particular, we study how the attack strength and the quality of the unlabeled data affect adversarial robustness in this framework. Our results show that with a high-quality unlabeled data generator, adversarial training can benefit greatly from this framework under large attack strength, while a poor generator can still help to some extent. To make adaptions concerning the quality of generated data, we propose an algorithm that performs online adjustment to the weight between the labeled real data and the generated data, aiming to optimize the adversarial risk. Numerical studies are conducted to verify our theories and show the effectiveness of the proposed algorithm.

## 1 Introduction

Adversarial training is a popular and simple way to improve the adversarial robustness of modern machine learning models. There are fruitful results in the theoretical justification and methodology development. Among various research directions, one interesting aspect is to use extra unlabeled data to assist adversarial training. Recent works successfully demonstrate a great improvement in the adversarial robustness with additional unlabeled data. For example, Carmon et al. (2019); Xing et al. (2021b, 2022), show that additional external real data help improve adversarial robustness; Gowal et al. (2021) uses synthetically generated data to improve the adversarial robustness and achieves the highest 66% adversarial testing accuracy for CIFAR-10 dataset under AutoAttack (AA) in Croce et al. (2020)[1] in the literature. The algorithms utilizing real/generated unlabeled data in adversarial training in Carmon et al. (2019); Gowal et al. (2021) are summarized in Figure 1. Note that to unify the these two algorithms, we view the unlabeled real data used in Carmon et al. (2019) as synthetic data generated from an *ideal generator*.

However, two fundamental questions about utilizing additional unlabeled data remain unclear.

First, as shown by Carmon et al. (2019), unlabeled data provide more information about the density of the data near the decision boundary, leading to a more robust adversarial estimator (e.g., it increases the robust test accuracy of CIFAR-10 from 53.08% to 59.53% for 8/255 $\mathcal{L}_\infty$ attack). However, the performance improvement of this training strategy is limited when the attack strength is small or zero. Therefore, it is natural to ask

*Q1: Compared to clean training, why and how can adversarial training significantly benefit from unlabeled data under the framework of Figure 1?*

---

[1] https://robustbench.github.io

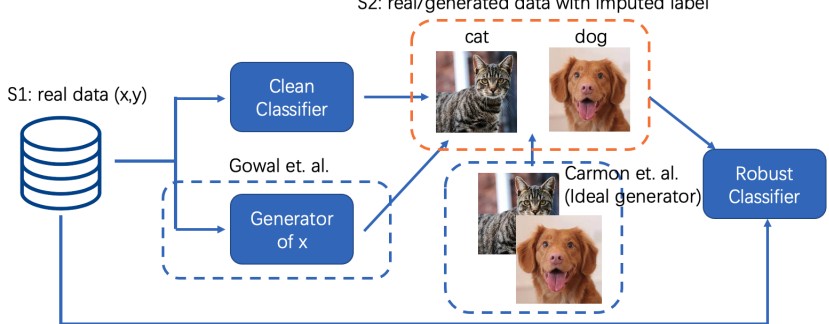

Figure 1: The procedure to do adversarial training with real/generated data. In Carmon et al. (2019), a clean classifier is trained to impute the label for additional real unlabeled data to form $S_2$. In Gowal et al. (2021), the clean classifier and the unlabeled data generator are trained via clean training to form $S_2$. Obtaining $S_2$, we use both $S_1$ and $S_2$ in the adversarial training. To unify the two algorithms, we call the unlabeled real data in Carmon et al. (2019) as data generated from *ideal generator*.

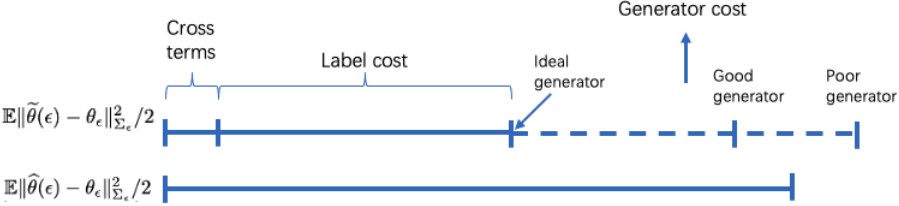

Figure 2: Exact risk decomposition of $\widetilde{\theta}(\epsilon)$ and $\widehat{\theta}(\epsilon)$ when $n_2 \to \infty$. The excess adversarial risk is $\mathbb{E}R(\theta, \epsilon) - R(\theta_\epsilon, \epsilon) = \mathbb{E}\|\theta - \theta_\epsilon\|_{\Sigma_\epsilon}^2/2 + o$ for $\theta \in \{\widehat{\theta}(\epsilon), \widetilde{\theta}(\epsilon)\}$ for some $\Sigma_\epsilon$. Ideal/good/poor generator refers to the unlabeled data generator ($\mathcal{P}_a$). If $n_2$ is finite, the label cost gets smaller because we use less noisy-label samples, but the generator cost gets larger because a smaller sample size gives larger estimation variance.

Second, besides the ideal generator in Carmon et al. (2019), Gowal et al. (2021) observe two other interesting phenomena regarding the quality of the generated unlabeled data. When the unlabeled data are generated from a generative model trained from original labeled data, although they introduce no more information beyond the original labeled data, the adversarial robustness gets enhanced. On the other hand, they also generate synthetic data from a multivariate Gaussian, which is much different from the real data. In this case, the generated data still slightly improve the robustness. Hence, it is important to investigate that

*Q2: How does the quality of the unlabeled data generator affect the adversarial robustness?*

This work aims to provide mathematical answers for the above two questions. Existing literature, (e.g., Alayrac et al., 2019; Uesato et al., 2019; Carmon et al., 2019; Zhai et al., 2019; Sehwag et al., 2021; Xing et al., 2021b, 2022) provide a justification that adversarial training potentially benefit from unlabeled/generated data and fail to explain that unlabeled data help little when attack strength is small. In contrast, we explicitly explain **why** and **how** adversarial training can be improved via an **exact** error decomposition rather than an upper bound statement.

For the convenience of our discussion below, we preliminarily introduce some notations, with detailed explanations given later. Denote $\widetilde{\theta}(\epsilon)$ as the robust estimator utilizing additional unlabeled data as in Figure 1, and $\widehat{\theta}(\epsilon)$ as the estimator via vanilla adversarial training. Denote the true distribution of $X$ as $\mathcal{P}_0$, and the distribution of the generated unlabeled data as $\mathcal{P}_a$. The true conditional distribution of $Y|X$ is $\mathcal{P}_y$, and $Y|X - \mathbb{E}[Y|X]$ has a distribution $\mathcal{P}_\epsilon$. As defined in Figure 1, $S_1$ is the set of original labeled training data with sample size $n_1$, and $S_2$ is the set of generated data with size $n_2$. Denote $\alpha = n_1/(n_1 + n_2)$ as the proportion of real data. Let $l_\epsilon$ be the adversarial loss function w.r.t. attack strength $\epsilon$, $l_0$ be the clean loss (i.e., $\epsilon = 0$), and $\theta_0$ be the parameter of the underlying true clean model.

**Contributions** We conduct theoretical analysis under some simple statistical setups. Through studying the convergence of $\widetilde{\theta}(\epsilon)$ and $\widehat{\theta}(\epsilon)$, we decompose the excess adversarial risk of $\widetilde{\theta}(\epsilon)$ into two major components (and their cross product): label cost (related to the quality of the imputed labels of generated unlabeled data) and generator cost (related to the quality of unlabeled data generator) as in Figure 2. Using this decomposition, we can answer *Q1* and *Q2*.

First, unlike $\widehat{\theta}(\epsilon)$ whose risk noticeably affected as $\epsilon$ increases, since the adversarial risk is stable to the imputation model, the label cost of $\widetilde{\theta}(\epsilon)$ is more insensitive w.r.t. $\epsilon$. In consequence, although in clean training, the label cost and the risk of $\widehat{\theta}(\epsilon)$ are comparable, the former one becomes relatively smaller as $\epsilon$ increases. As in Figure 2, when $\epsilon > 0$, the label cost of $\widetilde{\theta}(\epsilon)$ is smaller than the risk of $\widehat{\theta}(\epsilon)$. Using infinite data from the ideal unlabeled data generator, the generator cost becomes zero, and hence the overall cost of $\widetilde{\theta}(\epsilon)$ is smaller than $\widehat{\theta}(\epsilon)$.

Second, we characterize how the quality of the unlabeled data generator affects the final adversarial robustness. In general, a better data generator is always preferred. For the ideal generator, it will always improve adversarial robustness **effectively**, i.e., the generator cost is negligible when $n_2 \to \infty$. If the generator is learned from $S_1$ and captures extra information, it may **effectively** help as well when $n_2 \to \infty$. For a poor generator, it may **slightly** improve the performance for a large attack when $n_2$ is small, and the generator cost is always not negligible.

Finally, following Gowal et al. (2021), we balance the weights of $S_1$ and $S_2$ in the adversarial training process in order to achieve the optimal adversarial robustness. We show that a larger $n_2$ is always preferred with a proper choice of weight, even if generated from non-ideal generators. Tuning the best choice of weights via repeated trial-and-error runs can be infeasible due to the slow convergence of adversarial training. We propose an algorithm that dynamically adapts the weight during the training of neural networks and shows its promising performance in Section 5.

## 2 Adversarial Training

To formally introduce adversarial training, let $l_0$ denote the loss function and $f_\theta(x)$ be the model with parameter $\theta$. The adversarial loss function $l_\epsilon$ and the (population) adversarial risk are defined as

$$R(\theta, \epsilon) := \mathbb{E}_{\mathcal{P}_0} \left[ l_0 \left( x + A_\epsilon(f_\theta, x, y), y, \theta \right) \right] := \mathbb{E}_{\mathcal{P}_0} \left[ l_\epsilon \left( x, y, \theta \right) \right],$$

where $A_\epsilon$ is an attack of strength $\epsilon > 0$ and intends to deteriorate the loss in the following way

$$A_\epsilon(f_\theta, x, y) := \underset{z \in B_p(0, \epsilon)}{\operatorname{argmax}} \left\{ l_0(x + z, y, \theta) \right\}, \tag{1}$$

where $B_p(x, r)$ is a $\mathcal{L}_p$ ball centering at $x$ with radius $r$. Denote $\theta_\epsilon = \operatorname{argmin}_\theta R(\theta, \epsilon)$. Note that for simplicity, we only consider $p = 2$ in our theorems and simulations, while in the numerical studies of real-data experiment, we follow the routine setup and set $p = \infty$.

Given the data set $S_1$ consisting of $n_1$ i.i.d. labeled samples, the estimator $\widehat{\theta}(\epsilon)$ from the vanilla adversarial training aims to minimize the empirical version of $R(\theta, \epsilon)$:

$$\widehat{R}(\theta, \epsilon) = \frac{1}{n_1} \sum_{(x, y) \in S_1} l_\epsilon \left( x, y, \theta \right). \tag{2}$$

To take the advantage of the extra unlabeled data, we first assign pseudo-response variable $\widehat{y} = g(f_{\widehat{\theta}(0)}(x), \varepsilon)$ to them, where for regression, $g(a, b) = a + b$ and $\varepsilon$ follows the true noise distribution; for classification, $g(a, b) = 1(a \geq b)$ and $\varepsilon \sim \text{Unif}[0, 1]$. Then $\widetilde{\theta}(\epsilon)$ aims to minimize

$$\widetilde{R}(\theta, \epsilon) = \frac{1}{n_1 + n_2} \left( \sum_{(x, y) \in S_1} l_\epsilon \left( x, y, \theta \right) + \sum_{(x, \widehat{y}) \in S_2} l_\epsilon \left( x, \widehat{y}, \theta \right) \right). \tag{3}$$

To evaluate the performance of an estimator $\theta$, we use the excess adversarial risk $\mathbb{E}R(\theta, \epsilon) - R(\theta_\epsilon, \epsilon)$.

Besides the pseudolabel $\widehat{y}$, we define the the imaginary "true" response for the unlabeled data $y = g(f_{\theta_0}(x), \varepsilon)$, where $\varepsilon$ is the same realization used in generating pseudolabels. These true responses are not observable but will be used when analyzing the convergence rate of $\widetilde{\theta}(\epsilon)$. To evaluate the size of vectors and matrices, we denote $\| \cdot \|$ as the $\mathcal{L}_2$ norm of vectors, operator norm of matrices, and denote $\|a\|_A^2 := a^\top A a$ for any vector $a$ and positive definite matrix $A$.

# 3   Related Literature

Commonly used techniques (without extra data fed into training process) in adversarial training include adversarial regularization Goodfellow et al. (2015); Zhang et al. (2019); Wang et al. (2019b), curriculum-based algorithms Cai et al. (2018); Zhang et al. (2020a).

Theoretical investigations, besides the aforementioned works, have been conduct for adversarial training from different perspectives. For instance, Chen et al. (2020); Javanmard et al. (2020); Javanmard & Soltanolkotabi (2020); Taheri et al. (2021); Yin et al. (2018); Raghunathan et al. (2019); Najafi et al. (2019); Min et al. (2020); Hendrycks et al. (2019); Wu et al. (2020b); Xing et al. (2021a) studied the statistical properties of adversarial training, Sinha et al. (2017); Wang et al. (2019a) studied the optimization convergence of adversarial training, Zhang et al. (2020b); Allen-Zhu & Li (2020); Wu et al. (2020a); Xiao et al. (2021) studied theoretical issues related to adversarial training with Deep learning models.

# 4   Main Result

In this section, we present the main theorem, answer the two key questions, and show that balancing the weights of $S_1$ and $S_2$ leads to the better utility of the generated data. **Due to the page limit, we postpone the whole simulation study section to Appendix B.** Briefly speaking, our simulation results justify all the theoretical results below.

## 4.1   General Convergence Result

To present our theoretical investigations in $\widetilde{\theta}(\epsilon)$, we first introduce some assumptions:

**Assumption 1.** *The loss function satisfies the following conditions:*

- *A1. The clean loss function $l_0$ is the square loss or logistic loss. The attributes $X \in \mathbb{R}^d$ follow a sub-Gaussian distribution. The true clean model $\theta_0$ is defined as $\mathbb{E}[Y|X = x] = x^\top \theta_0$ for regression and $P(Y = 1|X = x) = 1/(1 + \exp(x^\top \theta_0))$ for classification. In addition, it satisfies that $\|\theta_0\| \leq b_0$ for some constant $b_0 > 0$ and the noise in regression has finite variance. The distribution $\mathcal{P}_a$ is sub-Gaussian as well.*

- *A2. The distribution $\mathcal{P}_a$ and $n_2$ satisfy $\|\mathbb{E}_{\mathcal{P}_a \otimes \mathcal{P}_y} \frac{\partial}{\partial \theta_\epsilon} l_\epsilon(X, Y, \theta_\epsilon)\| = o((n_1 + n_2)/(n_2 \log n_1))$, where $\mathcal{P}_a \otimes \mathcal{P}_y$ denotes the joint distribution induced by a marginal distribution $\mathcal{P}_a$ and a conditional distribution $\mathcal{P}_y$.*

Assumption A1 is for the simplicity of derivation. When doing Taylor expansion as in (4), the remainder term $o$ (if exists) does not explode under A1. Under Assumption A2, if $\mathcal{P}_a$ is of poor quality, i.e., $\|\mathbb{E}_{\mathcal{P}_a \otimes \mathcal{P}_y} \frac{\partial}{\partial \theta_\epsilon} l_\epsilon(X, Y, \theta_\epsilon)\|$ is not vanishing, then one can only take a relatively small $n_2$ compared to $n_1$. Vice versa, to allow a large $n_2$, the generator $\mathcal{P}_a$ need to be almost "unbiased". It is possible to relax Assumptions A1 and A2 as discussed in Section E in the appendix. However, the relaxations are technical and tailored to the derivations of Theorem 1.

The following theorem is a general result to decompose $\mathbb{E}R(\widetilde{\theta}(\epsilon), \epsilon) - R(\theta_\epsilon, \epsilon)$:

**Theorem 1.** *Under Assumption A1 to A2, assuming the unlabeled data generator is independent to $S_1$, when $n_1 \to \infty$, the excess adversarial risk using $\widetilde{\theta}(\epsilon)$ is*

$$\mathbb{E}R(\widetilde{\theta}(\epsilon), \epsilon) - R(\theta_\epsilon, \epsilon) = \frac{1}{2}\mathbb{E}\left\|\widetilde{\theta}(\epsilon) - \theta_\epsilon\right\|_{\Sigma_\epsilon}^2 + o, \tag{4}$$

*where $\theta_\epsilon - \widetilde{\theta}(\epsilon)$ is dominated by*

$$\underbrace{\frac{n_2 \Sigma_\epsilon^{-1}}{n_1 + n_2}\mathbb{E}_{\mathcal{P}_a \otimes \mathcal{P}_y}\left(\frac{\partial}{\partial \theta_\epsilon}l_\epsilon(X, \widehat{Y}, \theta_\epsilon) - \frac{\partial}{\partial \theta_\epsilon}l_\epsilon(X, Y, \theta_\epsilon)\right)}_{= \frac{n_2}{n_1 + n_2}\Sigma_\epsilon^{-1}\widetilde{\Sigma}_\epsilon(\widehat{\theta}(0) - \theta_0) + o := E_1} + \underbrace{\frac{\Sigma_\epsilon^{-1}}{n_1 + n_2}\left(\sum_{S_1, S_2}\frac{\partial}{\partial \theta_\epsilon}l_\epsilon(x, y, \theta_\epsilon)\right)}_{:= E_2},$$

*and*

$$\Sigma_\epsilon = \frac{\partial^2}{\partial \theta_\epsilon^2}\mathbb{E}_{\mathcal{P}_0 \otimes \mathcal{P}_y}l_\epsilon(X, Y, \theta_\epsilon), \ \widetilde{\Sigma}_\epsilon = \frac{\partial^2}{\partial \theta_\epsilon \partial \theta_0}\mathbb{E}_{\mathcal{P}_a}\mathbb{E}_\varepsilon l_\epsilon(X, g(f_{\theta_0}(X), \varepsilon), \theta_\epsilon).$$

*The term "o" represents the remainder terms which are not dominant.*

*Therefore, taking the decomposition of $\widetilde{\theta}(0)$ into $\mathbb{E}R(\widetilde{\theta}(\epsilon), \epsilon)$, the excess adversarial risk becomes*

$$\mathbb{E}R(\widetilde{\theta}(\epsilon), \epsilon) - R(\theta_\epsilon, \epsilon) = \frac{1}{2}\mathbb{E}_{\widehat{\theta}(0)}\|E_1\|^2_{\widetilde{\Sigma}_\epsilon} + \frac{1}{2}\mathbb{E}_{S_1,S_2}\|E_2\|^2_{\widetilde{\Sigma}_\epsilon} + \mathbb{E}_{S_1,S_2}E_1^\top \widetilde{\Sigma}_\epsilon E_2 + o,$$

*where $\mathbb{E}_{\widehat{\theta}(0)}\|E_1\|^2_{\widetilde{\Sigma}_\epsilon}/2$ quantifies how the quality of pseudolabel affects the excess risk (label cost), and $\mathbb{E}_{S_1,S_2}\|E_2\|^2_{\widetilde{\Sigma}_\epsilon}/2$ quantifies how the quality of $\mathcal{P}_a$ affects the excess risk (generator cost).*

*Meanwhile, the vanilla adversarial training estimate $\widehat{\theta}(\epsilon)$ satisfies*

$$\theta_\epsilon - \widehat{\theta}(\epsilon) = \Sigma_\epsilon^{-1}\left[\frac{1}{n_1}\frac{\partial}{\partial\theta_\epsilon}\left(\sum_{S_1} l_\epsilon(x,y,\theta_\epsilon)\right)\right] + o.$$

A proof sketch can be found in Section 6, while the detailed proof is postponed to the appendix. For simplicity of the derivation, $\mathcal{P}_a$ in Theorem 1 is independent to $S_1$. A case-by-case study is essential if $\mathcal{P}_a$ is trained from $S_1$. In Example 2 later, we provide a case where $\mathcal{P}_a$ is trained from $S_1$.

Based on Theorem 1, as shown in Figure 2 in the introduction, the error of $\widetilde{\theta}(\epsilon)$ consists of two parts and their cross terms:

**Label cost,** $\mathbb{E}_{\widehat{\theta}(0)}\|E_1\|^2_{\widetilde{\Sigma}_\epsilon}/2$: It is mainly due to the discrepancy between the pseudolabel $\widehat{y}$ and "true" label $y$ of the generated data. The label cost depends on both clean training error $\widehat{\theta}(0) - \theta_0$ and the matrix $\Sigma_\epsilon^{-1}\widetilde{\Sigma}_\epsilon$, and is maximized when $n_2 \to \infty$. It measures the quality of the pseudolabel.

**Generator cost,** $\mathbb{E}_{S_1,S_2}\|E_2\|^2_{\widetilde{\Sigma}_\epsilon}/2$: By the definition of $\theta_\epsilon$, i.e. the global minima of the adversarial risk, the expectation of $\frac{\partial}{\partial\theta_\epsilon}l_\epsilon(x,y,\theta_\epsilon)$ under $\mathcal{P}_0 \otimes \mathcal{P}_y$ is $\mathbf{0}$, thus $\sum_{S_1}\frac{\partial}{\partial\theta_\epsilon}l_\epsilon(x,y,\theta_\epsilon) = O_p(\sqrt{n_1})$. Consequently, the generator cost mostly rely on the term $\sum_{S_2}\frac{\partial}{\partial\theta_\epsilon}l_\epsilon(x,y,\theta_\epsilon)$, which depends on the quality of the unlabeled data generator $\mathcal{P}_a$. For ideal $\mathcal{P}_a$, i.e., $\mathcal{P}_a = \mathcal{P}_0$, $\mathbb{E}_{\mathcal{P}_a \otimes \mathcal{P}_y}\frac{\partial}{\partial\theta_\epsilon}l_\epsilon(X,Y,\theta_\epsilon) = \mathbf{0}$, and the generator cost goes to zero when $n_2 \to \infty$. For non-ideal $\mathcal{P}_a$, $\|\mathbb{E}_{\mathcal{P}_a \otimes \mathcal{P}_y}\frac{\partial}{\partial\theta_\epsilon}l_\epsilon(X,Y,\theta_\epsilon)\|$ may be vanishing, but not exact zero.

Besides Figure 2, we also provide graphical illustration on how the label cost and generator cost change along $n_2/n_1$ in Section F to ease the understanding.

## 4.2 Discrepancy between Clean and Adversarial Training with Unlabeled Data

We provide some intuitions to explain the discrepancy in the algorithm performance between $\epsilon = 0$ and $\epsilon > 0$ from two aspects, the loss function aspect and information (minimax lower bound) aspect. The following discussion assumes that the ideal data generator is used.

**Loss function and label cost** We rewrite the adversarial loss as $R_\epsilon(\theta, \theta_0) = \mathbb{E}l_\epsilon(X, g(f_{\theta_0}(X), \varepsilon), \theta)$ to explicitly reveal its dependency on the true parameter $\theta_0$. A crucial difference between clean training and adversarial training is the roles of the $\theta$ and $\theta_0$ in the loss functions. For clean loss, e.g. linear regression, the loss function can be rewritten as $(x^\top\theta - x^\top\theta_0 - \varepsilon)^2$, where $\theta$ and $\theta_0$ play symmetric roles. In contrast, for adversarial loss, the loss becomes $((x + A_\epsilon(f_\theta, x, y))^\top\theta - x^\top\theta_0 - \varepsilon)^2$, thus $\theta$ and $\theta_0$ have asymmetric roles. The adversarial loss is more sensitive to $\theta$ than $\theta_0$. Consequently, when $\widehat{\theta}(0)$ is treated as true parameter and used to impute labels for $S_2$, the impact of the error in $\widehat{\theta}(0)$ is much less influential for adversarial training than clean training.

In Theorem 1, the matrix $\widetilde{\Sigma}_\epsilon$ reflects the sensitivity mentioned in the above illustration. Assume $\epsilon = 0$, for both

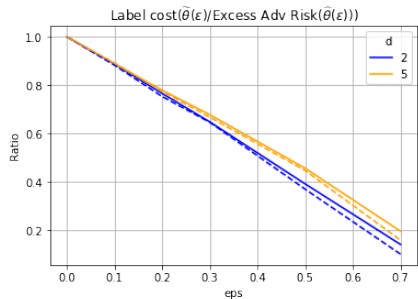

Figure 3: The ratio between label cost of $\widetilde{\theta}(\epsilon)$ and $\mathbb{E}\|\widehat{\theta}(\epsilon) - \theta_\epsilon\|^2_{\widetilde{\Sigma}_\epsilon}$ in linear regression, $n_2 \to \infty$. Solid line: theory. Dashed line: simulation. Derivations are in Appendix H.

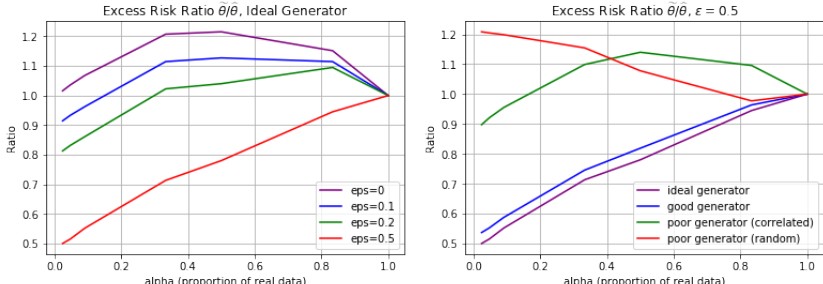

Figure 4: Simulation: how the attack strength ($\epsilon$), quality of data generator, and the proportion of real data ($\alpha = n_1/(n_1+n_2)$) affect the convergence of the adversarial estimator. The y-axis represents the excess adversarial risk ratio of $\widetilde{\theta}(\epsilon)$ and $\widehat{\theta}(\epsilon)$, i.e., $[\mathbb{E}R(\widetilde{\theta}(\epsilon), \epsilon) - R(\theta_\epsilon, \epsilon)]/[\mathbb{E}R(\widehat{\theta}(\epsilon), \epsilon) - R(\theta_\epsilon, \epsilon)]$. A smaller-than-one ratio implies better performance of $\widetilde{\theta}(\epsilon)$. Detailed model information is in Appendix B: the left panel corresponds to the left panel of Figure B.1, and the right panel is for $\epsilon = 0.5$ from four different generators as in Figure B.1 and B.2.

linear regression and logistic regression, we have $-\widetilde{\Sigma}_0 = \Sigma_0$, so when $n_2 \to \infty$,

$$\widetilde{\theta}(0) - \theta_0 = \widehat{\theta}(0) - \theta_0 + o,$$

which means that clean training cannot be improved using $S_2$ with pseudolabel.

On the other hand, for adversarial training, the matrix $\Sigma_\epsilon^{-1}\widetilde{\Sigma}_\epsilon$ is no longer $-I_d$, and the singular values get changed. As a consequence, the label cost is cheaper than the overall risk of $\widehat{\theta}(\epsilon)$ even if the label cost is maximized as $n_2 \to \infty$. A toy simulation in Figure 3 justifies this. In Figure 3, we calculate the theoretical values of the label cost of $\widetilde{\theta}(\epsilon)$ and the risk of $\widehat{\theta}(\epsilon)$ to compare them. When $\epsilon$ gets larger, the label cost in $\widetilde{\theta}(\epsilon)$ becomes much smaller compared to the overall risk of $\widehat{\theta}(\epsilon)$.

Together with an ideal $\mathcal{P}_a$ that yields zero generator cost ($n_2 \to \infty$), we obtain that $\widetilde{\theta}(\epsilon)$ is much better than $\widehat{\theta}(\epsilon)$. The simulation results in the left panel of Figure 4 verifies this as well. All these theoretical insights and numerical observations under $\epsilon > 0$, are summarized in Figure 2. Figures 2 and 3 jointly are the main **answer towards Q1**. Note that we assert adversarial training benefits much more from unlabeled data than clean training, rather than that clean training cannot benefit from unlabeled data. See the remark below:

**Remark 1.** *It is possible to improve clean training using unlabeled data with pseudolabels. In Lee et al. (2013); Sun et al. (2017), in each training iteration, pseudolabels are updated based on the model of the current step. This is equivalent to a SGD with the gradient of $\sum_{S_1} l(f_\theta(x), y)/n_1 + \mathcal{L}(\theta, \theta^{(t)})$ for some regularization term $\mathcal{L}$ that stabilizes the training process, where $\theta^{(t)}$ as the current model.*

**Minimax lower bound** Besides the exact decomposition of $\widetilde{\theta}(\epsilon)$ to answer *Q1*, we also provide minimiax lower bound result to reveal another difference between clean and adversarial training.

Minimax rate refers to the best possible convergence rate that can be achieved by **any** estimator in the worst case given finite samples. As mentioned in Dan et al. (2020), it represents the information-theoretical limit given the data set.

In general, for the clean estimate without model misspecification, this information limit regards mostly the uncertainty about the conditional distribution of $Y$ given $X = x$, which completely determines $\theta_0$. The additional information about the marginal distribution of $X$ only affects the multiplicative constant of the bound but not the rate of convergence.

However, the true robust model $\theta_\epsilon$ can depend on both $Y|X$ and $X$ for the adversarial estimate. The following example demonstrates the difference between clean and adversarial training:

**Example 1.** *Consider linear regression under Gaussian model $Y = \theta_0^\top X + \varepsilon$ where $X \sim N(\mathbf{0}, \Sigma)$ and $\varepsilon \sim N(0, \sigma^2)$. Then $\theta_0$ is the best clean model, and $\theta_\epsilon = (I + \lambda_\epsilon \Sigma^{-1})^{-1}\theta_0$ for some $\lambda_\epsilon > 0$.*

The derivation of Example 1 can be found in Appendix H. Based on Example 1, it is intuitive that the minimax lower bound for the adversarially robust estimate is contributed by two sources, one for the

uncertainty of $\theta_0$, and the other one for the distribution of $X$, where the additional unlabeled data can improve the latter term. The following proposition reveals this difference theoretically:

**Proposition 1.** *Under the setup of Example 1, assume $\|\theta_0\| \leq b_0$ and there are $n_2$ unlabeled real data. Assume $R(\theta_\epsilon, \epsilon)$ is bounded from above and below, and $\epsilon \in [0, \epsilon^*]$ with some constant $\epsilon^*$ for $\mathcal{L}_2$ attack. Assume $\Sigma$ is unknown. Then for some positive constant $c$, for any estimator $\widehat{f}$,*

$$\inf_{\widehat{f}} \sup_{\mathcal{P}_0 \otimes \mathcal{P}_y} \mathbb{E}[R(\widehat{f}, \epsilon) - R(\theta_\epsilon, \epsilon)] \geq c \max \left( \inf_{\widehat{\theta}_0} \sup_{\theta_0} \mathbb{E}\|\widehat{\theta}_0 - \theta_0\|^2, \epsilon^2 b_0^2 \inf_{\widehat{\Sigma}} \sup_{\Sigma} \mathbb{E}\|\widehat{\Sigma} - \Sigma\|^2 \right).$$

### 4.3 Quality of Unlabeled Data Matters

We study how the quality of a non-ideal unlabeled data generator affects adversarial training. Unsurprisingly, we always prefer a high-quality/ideal $\mathcal{P}_a$. In addition, when there is some extra information of the marginal distribution of $X$ (such as parametric modeling assumptions) which cannot be directly utilized in the vanilla adversarial training process, training $\mathcal{P}_a$ from $S_1$ can potentially give a high-quality adversarially robust model. On the other hand, given a poor $\mathcal{P}_a$, via bias-variance trade-off, it is still possible to reduce the error of $\widetilde{\theta}(\epsilon)$. Note that the discussion in the section assumes that the unlabeled data generator ($\mathcal{P}_a$) is independent to the training data $S_1$, as assumed in Theorem 1, unless stated otherwise.

**A high-quality $\mathcal{P}_a$ is preferred**  The term $E_2$ in Theorem 1 satisfies

$$\mathbb{E}E_2 E_2^\top = \underbrace{\frac{n_2^2}{(n_1 + n_2)^2} \left[ \mathbb{E}_{\mathcal{P}_a \otimes \mathcal{P}_y} \frac{\partial}{\partial \theta_\epsilon} l_\epsilon(X, Y, \theta_\epsilon) \right] \left[ \mathbb{E}_{\mathcal{P}_a \otimes \mathcal{P}_y} \frac{\partial}{\partial \theta_\epsilon} l_\epsilon(X, Y, \theta_\epsilon) \right]^\top}_{\text{Square bias}} \tag{5}$$

$$+ \underbrace{\frac{n_1}{(n_1 + n_2)^2} Var_{\mathcal{P}_0 \otimes \mathcal{P}_y} \left( \frac{\partial}{\partial \theta_\epsilon} l_\epsilon(X, Y, \theta_\epsilon) \right) + \frac{n_2}{(n_1 + n_2)^2} Var_{\mathcal{P}_a \otimes \mathcal{P}_y} \left( \frac{\partial}{\partial \theta_\epsilon} l_\epsilon(X, Y, \theta_\epsilon) \right)}_{\text{Variance}}.$$

For the variance term in (5), when $n_2 \to \infty$, it is always negligible for reasonable $\mathcal{P}_a$.

For the square bias in (5), it is in general nonzero if $\mathcal{P}_a \neq \mathcal{P}_0$, and is similar to the maximum mean discrepancy (MMD, Gretton et al., 2012) that measures the dissimilarity between $\mathcal{P}_a$ and $\mathcal{P}_0$. There exists some function class $\mathcal{F}$ such that $\partial l_\epsilon / \partial \theta_\epsilon \in \mathcal{F}$ and

$$\left\| \mathbb{E}_{\mathcal{P}_a \otimes \mathcal{P}_y} \frac{\partial}{\partial \theta_\epsilon} l_\epsilon(X, Y, \theta_\epsilon) \right\| = \left\| \mathbb{E}_{\mathcal{P}_a \otimes \mathcal{P}_y} \frac{\partial}{\partial \theta_\epsilon} l_\epsilon(X, Y, \theta_\epsilon) - \underbrace{\mathbb{E}_{\mathcal{P}_0 \otimes \mathcal{P}_y} \frac{\partial}{\partial \theta_\epsilon} l_\epsilon(X, Y, \theta_\epsilon)}_{=0} \right\|$$

$$\leq \sqrt{d} \text{MMD}(\mathcal{F}, \mathcal{P}_a, \mathcal{P}_0).$$

Based on the above decomposition, a higher quality of the unlabeled data generator is preferred because it leads to small generator cost. And as shown in Figure 2, with the ideal $\mathcal{P}_a$, the generator cost can be zero when $n_2 \to \infty$. Using an ideal/good $\mathcal{P}_a$, we can have $\mathbb{E}\|\widetilde{\theta}(\epsilon) - \theta_\epsilon\|_{\Sigma_\epsilon}^2 < \mathbb{E}\|\widehat{\theta}(\epsilon) - \theta_\epsilon\|_{\Sigma_\epsilon}^2$.

A graphical illustration can be found in the right panel of Figure 4. When $\epsilon$ is large, we know that the ideal $\mathcal{P}_a$ will efficiently improve the adversarial robustness. When $\mathcal{P}_a$ gets worse, there is less improvement in the adversarial robustness. This is the main **answer towards Q2**, and we illustrate some specific cases in the following discussions.

**Training data generator from $S_1$**  For the vanilla adversarial training, there is no trivial way to incorporate the extra information of the marginal distribution of $X$ into the optimization of (2). In contrast, $\widetilde{\theta}(\epsilon)$ can use the extra information to train a good unlabeled data generator from $S_1$.

**Example 2** (Sparse Covariance Matrix Estimate). *Assume $X \sim N(\mathbf{0}, \Sigma)$ and $\Sigma$ is unknown.*

*We follow Cai et al. (2010) to consider a family of sparse covariance matrix as follows:*

$$\mathcal{F}_\beta = \left\{ \Sigma : \max_j \sum_{\{i : |i-j| > k\}} |\sigma_{ij}| \leq M k^{-\beta} \; \forall k, \; \lambda_{\max}(\Sigma) \leq M_0, \; \lambda_{\min}(\Sigma) \geq m_0 > 0 \right\}.$$

*If we use the sparse estimator proposed in Cai et al. (2010), then $\mathbb{E}\|\widehat{\Sigma}_{sparse} - \Sigma\|^2 = O(n_1^{-2\beta/(2\beta+1)})$.*

Besides, when $\widehat{\Sigma}$ is either the sample covariance matrix or $\widehat{\Sigma}_{sparse}$, with probability tending to 1 over the generation of $S_1$, the decomposition of $\widetilde{\theta}(\epsilon)$ in Theorem 1 still holds, and the bias in (5) satisfies

$$\frac{n_2^2}{(n_1+n_2)^2} \left\| \mathbb{E}_{\mathcal{P}_a \otimes \mathcal{P}_y} \frac{\partial}{\partial \theta_\epsilon} l_\epsilon(X, Y, \theta_\epsilon) \right\|^2 = O\left( \frac{n_2^2 \|\widehat{\Sigma} - \Sigma\|^2}{(n_1+n_2)^2} \right). \tag{6}$$

If $\Sigma \in \mathcal{F}_\beta$ and $d \gg n^{1/(2\beta+1)}$, when $n_2 \to \infty$, $\widetilde{\theta}(\epsilon)$ with $\widehat{\Sigma}_{sparse}$ leads to a rate of $o(d/n_1)$ for the generator cost, which is negligible compared to the label cost.

Compared to Theorem 1, a difference in the assumption is that in Example 2, the generator is not independent to $S_1$. Therefore, the proof of (6), presented in Appendix G, is more involved.

**Bias-variance trade-off in poor $\mathcal{P}_a$** For a poor unlabeled data generator, when $n_2 \to \infty$, it leads to a poor $\widetilde{\theta}(\epsilon)$ as shown in Figure 2. However, based on the bias-variance trade-off of $E_2$, for a relatively $n_2$, it is still possible to obtain a $\widetilde{\theta}(\epsilon)$ better than $\widehat{\theta}(\epsilon)$.

**Proposition 2** (Bias-variance trade-off). *Assume the conditions of Theorem 1 hold, $n_2 \ll n_1$ and the unlabeled data generator is independent to $S_1$, then*

$$\|\mathbb{E}\widetilde{\theta}(\epsilon) - \theta_\epsilon\|_{\Sigma_\epsilon}^2 = O\left( \frac{dn_2^2}{(n_1+n_2)^2} \right), \quad tr(Var[\Sigma_\epsilon^{1/2}(\widetilde{\theta}(\epsilon) - \theta_\epsilon)]) = O\left( \frac{d}{n_1+n_2} \right).$$

As shown by Proposition 2, there is a trade-off between bias and variance w.r.t. the size of $S_2$. This implies a U-shaped curve of the generator cost w.r.t. $S_2$, and there exists some small $n_2$ that strikes the balance of this trade-off and can improve the adversarial robustness.

## 4.4 Balancing the Weights between $S_1$ and $S_2$

Following Gowal et al. (2021) and Carmon et al. (2019), we try to improve the performance of $\widetilde{\theta}(\epsilon)$ via balancing the weights of $S_1$ and $S_2$ in (3), and consider the weighted minimization,

$$\widetilde{R}(\theta, w, \epsilon) = \frac{1}{n_1 + wn_2} \left( \sum_{(x,y)\in S_1} l_\epsilon(x, y, \theta) + w \sum_{(x,\widehat{y})\in S_2} l_\epsilon(x, \widehat{y}, \theta) \right). \tag{7}$$

When using the ideal $\mathcal{P}_a$ and assuming $\widehat{\theta}(0) \equiv \theta_0$ (i.e., we can generate extra independent labeled real data), trivially the optimal $w$ minimizing the risk of $\widetilde{\theta}(\epsilon)$ is 1. When using non-ideal $\mathcal{P}_a$ or $\widehat{\theta}(0) \neq \theta_0$, intuitively, one need to downweight the generated data and the best $w < 1$. We demonstrate this observation in the simulation in Figure B.3 as well.

In addition, as stated in the following proposition, through taking the optimal choice of $w$, for any $\mathcal{P}_a$, a larger $n_2$ always leads to better $\widetilde{\theta}$ when $\epsilon > 0$.

**Proposition 3.** *We assume the assumptions of Theorem 1 hold, and consider a fixed $(\epsilon > 0, n_1)$. Denote $\widetilde{\theta}(w, n_2)$ as the minimizer of (7), and $w^*(n_2)$ minimizes the excess adversarial risk of $\widetilde{\theta}(w, n_2)$ w.r.t. $w$. Then for $n_2' > n_2$, taking $w'$ such that $w'n_2' = w^*(n_2)n_2$, the risk of $\widetilde{\theta}(w', n_2')$ is smaller than $\widetilde{\theta}(w^*(n_2), n_2)$, and the risk of $\widetilde{\theta}(w^*(n_2'), n_2')$ is further smaller than $\widetilde{\theta}(w', n_2')$. As a result, a larger $n_2$ always gives a better $\widetilde{\theta}$.*

## 4.5 Summary of Numerical Experiments

**Due to the space limit, we postpone simulations and most real experiments to Appendix B, C.** Below is a summary of simulation and empirical studies.

In Appendix B, we aim to use simple models to numerically verify: (1) given the ideal data generator, the performance of $\widetilde{\theta}(\epsilon)$ is better than $\widehat{\theta}(\epsilon)$ when $\epsilon$ deviates from zero; (2) the better quality of the data generator implies the better performance of $\widetilde{\theta}(\epsilon)$; and (3) balancing the weight between $S_1$ and $S_2$ improves the performance. We observe all (1) to (3) in the simulations.

In Appendix C, we aim to verify that the label cost and the generator cost are important factors in deep learning. We aim to show (1) adding more unlabeled samples from the ideal generator will improve adversarial robustness, and (2) adding unlabeled samples from a poor generator with a small $n_2$ will slightly improve the performance. We perform an experiment on the CIFAR-10 data set. We take a part of the samples as labeled training data and remove the label of the other samples. Therefore, if we train a classifier to classify airplanes and cars, then (i) the unlabeled airplane and car pictures can be viewed as ideal unlabeled samples, and (ii) the unlabeled samples from other classes are viewed as from a poor generator. Our empirical results verify (1) and (2). Besides, we also find that the unlabeled data with the pseudo label can improve the adversarial training as much as labeled data of the same sample size (the middle panel of Figure C.1). This observation further verifies the effectiveness of unlabeled data and implies that the label cost is small in adversarial training.

## 5   How to Decide a Proper $w$

To determine a proper $w$, it is possible to conduct cross validation for simple models. However, due to the heavy computation of adversarial training under DNN models, this is infeasible for real applications. Our proposed algorithm in below aims to simplify the tuning process of $w$.

To design a proper metric to tune $w$, denote $b$ as the average adversarial loss difference in $S_1$ and $S_2$, $v_1$ and $v_2$ as the variance of the adversarial loss for $S_1$ and $S_2$ respectively. Since in real practice one can often obtains a high-quality clean classifier (i.e., low label cost), we tune $w$ based on the generator cost only. Inspired by the variance-bias trade-off in (5), we define a surrogate of the generator loss as

$$\kappa(b, v_1, v_2, n_1, n_2, w, w_{\text{bias}}) = w_{\text{bias}} \frac{b^2(wn_2)^2}{(n_1 + wn_2)^2} + \frac{v_1 n_1 + w^2 v_1 n_2}{(n_1 + wn_2)^2}.$$

Due to over-parameterization, the mean and variance of the gradient are volatile in neural networks. Hence, instead of using the gradient of loss (i.e., $\partial l_\epsilon / \partial \theta_\epsilon$) as suggested by (5), we use the adversarial loss to construct $\kappa$ and introduce $w_{\text{bias}}$ to balance $b$ and $(v_1, v_2)$. To control over-fitting problem, we tune $w$ during the first 20% iterations in the experiment. The algorithm is shown in Algorithm 1.

---

**Algorithm 1** Select $\alpha$ during Training

---

**Input:** Training dataset $S_1$ and $S_2$ with size $n_1$ and $n_2$, optimizer OPT, total number of iterations $T$, number of iteration per weight $\tau$, initial weight $w_0$, bias weight $w_{\text{bias}}$, decay factor $\delta$, number of epochs $K$ to train weight.
Initialize the model parameters $\theta$, take $w = w_0$.
**for** $k = 1, \ldots, T/\tau$ **do**
    $b = v_1 = v_2 = 0$.
    **for** $t = 1, \ldots, \tau$ **do**
        Calculate the mean and variance of adversarial loss for the samples from $S_1$ and $S_2$ in this batch as $b_{t1}, b_{t2}, v_{t1}, v_{t2}$.
        Update $b = b * \delta + b_{t2} - b_{t1}$, $v_1 = v_1 * \delta + v_{t1}$, $v_2 = v_2 * \delta + v_{t2}$.
        Update $\theta$ using OPT.
    **end for**
    Update $b = b/(1 - \delta)$, $v_1 = v_1/(1 - \delta)$, $v_2 = v_2/(1 - \delta)$.
    **if** $k \leq K$ **then**
        Update $w$ to reduce $\kappa$.
    **end if**
**end for**
**Output:** $\theta$.

---

In the experiment, we consider binary classification for the CIFAR-10 dataset to classify airplane and car. The implementation for all real-data experiments is modified from Rice et al. (2020)[2]. We take 500 samples from each class as labeled data, i.e., $n_1 = 1,000$. To form $S_2$, we sample $n_2/2$ data from the other 9,000 airplane and car pictures, and $n_2/2$ data from other classes, i.e., the unlabeled data generator generates both ideal and poor samples. The experiment setups are postponed to Appendix C. We repeat the experiment for 10 times to get the average robust testing accuracy and its standard error. The details of how to tune $w_{\text{bias}}$ and how to update $w$ using $\kappa$ are in Appendix C.

---

[2] https://github.com/locuslab/robust_overfitting

The results are summarized in Table 1. One can see that adjusting $w$ using Algorithm 1 can always lead to a better performance than the unweighted training and is as good as the best fixed $w$ obtained by grid search. We found that 75% of the total epochs are needed for each fixed $w$ to reveal the performance difference among $w$'s (Figure C.3), leading to long computing time for grid search of $w$. In contrast, our proposed method only needs several epochs to tune $w_{\text{bias}}$.

Table 1: Adversarial test accuracy using unweighted training ($w = 1$), Algorithm 1, and best fixed $w$. The latter two methods reaches similar performance.

| $n_2$ | 200 | 2000 | 4000 | 18000 |
|---|---|---|---|---|
| $w = 1$ | 0.7847(0.0041) | 0.7612(0.0168) | 0.8170(0.0096) | 0.8622(0.0057) |
| Algorithm 1 | 0.7870(0.0090) | 0.7942(0.0126) | 0.8407(0.0070) | 0.8765(0.0038) |
| Best fixed $w$ | 0.7868(0.0091) | 0.7957(0.0128) | 0.8429(0.0068) | 0.8754(0.0027) |

We also evaluate the performance of Algorithm 1 using the full CIFAR-10, CIFAR-100, and SVHN with the additional data generated by Gowal et al. (2021). The results are postponed to Appendix C.

## 6 Proof Sketch of Theorem 1

We assume $\widetilde{\theta}(\epsilon)$ is consistent to $\theta_\epsilon$ and start from the first-order optimality condition of $\widetilde{\theta}(\epsilon)$ and obtain that

$$
\begin{aligned}
\mathbf{0} &= \frac{1}{n_1 + n_2} \frac{\partial}{\partial \theta_\epsilon} \left( \sum_{S_1} l_\epsilon(x, y, \widetilde{\theta}(\epsilon)) + \sum_{S_2} l_\epsilon(x, \widehat{y}, \widetilde{\theta}(\epsilon)) \right) \qquad (8) \\
&= \underbrace{\frac{1}{n_1 + n_2} \frac{\partial}{\partial \theta_\epsilon} \left( \sum_{S_1} l_\epsilon(x, y, \widetilde{\theta}(\epsilon)) \right) - \frac{1}{n_1 + n_2} \frac{\partial}{\partial \theta_\epsilon} \left( \sum_{S_1} l_\epsilon(x, y, \theta_\epsilon) \right)}_{:=A_1} \\
&+ \underbrace{\frac{1}{n_1 + n_2} \frac{\partial}{\partial \theta_\epsilon} \left( \sum_{S_2} l_\epsilon(x, \widehat{y}, \widetilde{\theta}(\epsilon)) \right) - \frac{1}{n_1 + n_2} \frac{\partial}{\partial \theta_\epsilon} \left( \sum_{S_2} l_\epsilon(x, y, \widetilde{\theta}(\epsilon)) \right)}_{:=A_2} \\
&+ \underbrace{\frac{1}{n_1 + n_2} \frac{\partial}{\partial \theta_\epsilon} \left( \sum_{S_2} l_\epsilon(x, y, \widetilde{\theta}(\epsilon)) \right) - \frac{1}{n_1 + n_2} \frac{\partial}{\partial \theta_\epsilon} \left( \sum_{S_2} l_\epsilon(x, y, \theta_\epsilon) \right)}_{:=A_3} \\
&+ \frac{1}{n_1 + n_2} \frac{\partial}{\partial \theta_\epsilon} \left( \sum_{S_1} l_\epsilon(x, y, \theta_\epsilon) + \sum_{S_2} l_\epsilon(x, y, \theta_\epsilon) \right).
\end{aligned}
$$

We observe that, when $\widetilde{\theta}(\epsilon)$ is consistent, i.e., $\|\widetilde{\theta}(\epsilon) - \theta_\epsilon\| \xrightarrow{P} 0$, with probability tending to 1,

$$
\begin{aligned}
A_1 + A_3 &= \Sigma_\epsilon(\widetilde{\theta}(\epsilon) - \theta_\epsilon) + o, \\
A_2 &= \frac{n_2}{n_1 + n_2} \widetilde{\Sigma}_\epsilon(\widehat{\theta}(0) - \theta_0) + o.
\end{aligned}
$$

As a result, rearranging the terms in (8), we can prove Theorem 1. One the other hand, one can also show that $\|\widetilde{\theta}(\epsilon) - \theta_\epsilon\| \xrightarrow{P} 0$ because $\widetilde{R}(\theta, \epsilon) \to R(\theta, \epsilon)$ for any reasonable $\theta$.

## 7 Conclusion

This paper studies how adversarial training benefits from unlabeled (generated) data. We show that (i) the label cost of $\widetilde{\theta}(\epsilon)$ is small in adversarial training; (ii) with a high-quality data generator, a large $S_2$ leads to a negligible generator cost. These two facts together indicate that adversarial training benefits a lot from unlabeled (generated) data. Motivated by these observations and our theory, we balance the weights between $S_1$ and $S_2$ to improve the performance and propose an algorithm to determine the weight automatically during training.

## Acknowledgements

This project is partially supported by NSF-SCALE MoDL (2134209) and ONR N00014-22-1-2680.

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
