# OpenReview forum: "Why Do Artificially Generated Data Help Adversarial Robustness"
_NeurIPS.cc/2022/Conference — NeurIPS 2022 Accept_

### Official Review · Reviewer_VSRZ · 2022-06-15

**Rating:** 8
**Confidence:** 4
**Soundness:** 4 excellent
**Presentation:** 4 excellent
**Contribution:** 4 excellent

**Summary:**

In this paper, the authors provide a statistical insight to explain the reason that using unlabeled generated data can improve model's robustness. The experiments verify that the theories in this paper are correct and can enhance the model's robustness.

**Questions:**

1. In Figure 2, the authors suppose that $n_2 \to \infty$. Later, the authors analyze the case where $n_2$ is finite. I think there may exist a gap. Could the authors add more details for different ratio of $\frac{n_1}{n_2}$, and analyze the remainder term $o$?

2. What does ``unbiased'' mean in Line 127? I guess the authors judge a generator based on the gap between real data distribution and a data distribution generated by a generator. So, an ideal generator means it can generate a real data distribution. But, how do we define a good one and a poor one without any threshold?

**Limitations:**

I do not see any limitations.

**Strengths And Weaknesses:**

Strengths:

1. The writing is good. This paper is very easy to follow.

2. This paper gives detailed theoretical analysis of the reason that using unlabeled data can improve model's robustness.

3. The experimental results strongly verify the analysis is correct.

Weaknesses:

1. As the quality of unlabeled data matters, how to choose a proper generator for adversarial training is vital. However, the theory in this paper cannot directly guide people to train or choose such a generator. On the other hand, it seems like judging a poor generator is much easier. Could the authors attempt to solve this challenge?

---

> ### Author Response · Authors · 2022-08-01
> **Response**
>
> We appreciate your effort in reviewing our paper! There are some common updates in the paper that mentioned at the top comment, and below are answers for your questions:
>
> 1. Weakness, how to train a good data generator: We appreciate you sharing this question with us! We agree that studying how to improve the generator quality is vital.
>
>     * In our Example 2, we are showing how to better estimate the data distribution if $X\sim (\textbf{0},\Sigma)$ for some unknown $\Sigma$. With proper model assumptions, it is possible to improve the generator quality for simple models.
>
>     * In terms of real practice, it is still an open question of how to generate better synthetic data. In [1], they try diffusion models and GAN models, and DDPM shows the best performance among all the models in their paper.
>
>       In [1], besides comparing different generation models, they use different evaluation criteria to evaluate the performance of different models and how they affect the final adversarial training performance. For example, they consider coverage and complementarity and show that these criteria are related to adversarial training performance. Besides the metrics in [1], there are some other criteria, e.g., in [2].
>
>       Once the relationship between the evaluation metrics for  generators and adversarial training performance is well established, one can strive for better data synthesis under those metrics. This will be helpful to improve adversarial performance and, more importantly, can apply to many other applications for different robustness needs.
>
> 2. Q1, $n_2$ in Figure 2: Thank you for your useful suggestion! Figure 2 aims to explain the most important observations in the label cost and the generator cost, which are most clear when $n_2\rightarrow\infty$. In our revision, we provide two additional figures similar to Figure 2 to show how $n_2/n_1$ affects the performance under ideal/poor (independent of $S_1$) generators in Section E. In general,
>
>     * When $n_2/n_1\rightarrow\infty$, the label cost is minimized, and the generator cost only depends on the generator quality.
>
>     * When $n_2/n_1\rightarrow 0$, the sum of the label cost and the generator cost gets slightly decreased due to a bias-variance trade-off compared to $n_2=0$.
>
>     * When $n_2/n_1$ is finite and away from zero, because the generator cost and the label cost are both related to $n_2$ and are in the same order, it is hard to describe exactly how the sum of the costs is changed. But in general, with an ideal generator, a larger $n_2$ could lead to better adversarial robustness. Our simulations also verify this.
>
> 3. Q1, the small order term $o$: We use many $o$ in our derivations and formulas such that the representation of our result focuses on the most important term. In general, for consistent $\widetilde{\theta}(\epsilon)$ and $\widehat\theta(\epsilon)$, the $o$ term is always a negligible term.
>
> 4. Q2, "unbiased": In line 127, the word ``unbiased" refers to the scenario in Assumption A2, i.e., an asymptotically "unbiased" generator satisfies that $\|\mathbb{E}_{\mathcal{P}_a\otimes\mathcal{P}_y} \partial/\partial\theta_\epsilon l_\epsilon(X,Y,\theta_\epsilon)\|=o(1)$. Thanks for pointing out this issue. We updated the statement in our revision.
>
> References:
>
> [1] Sven Gowal, Sylvestre-Alvise Rebuffi, Olivia Wiles, Florian Stimberg, Dan Andrei Calian, and Timothy A Mann. Improving robustness using generated data. Advances in Neural Information Processing Systems, 34, 2021.
>
> [2] Ahmed Alaa, Boris Van Breugel, Evgeny S Saveliev, and Mihaela van der Schaar. How faithful is your synthetic data? sample-level metrics for evaluating and auditing generative models. In International Conference on Machine Learning, pages 290–306. PMLR, 2022.

---

### Official Review · Reviewer_bXme · 2022-06-19

**Rating:** 7
**Confidence:** 2
**Soundness:** 3 good
**Presentation:** 3 good
**Contribution:** 3 good

**Summary:**

This paper provides statistical insights to explain why the artificially generated data improve adversarial training. In particular, it studies how the attack strength and the quality of the unlabeled data affect adversarial robustness in the adversarial training framework of Carmon et al. (2019); Gowal et al. (2021). The results show that with a high-quality unlabeled data generator, adversarial training can benefit greatly from this framework under large attack strength, while a poor generator can still help to some extent. It then proposes an algorithm that performs online adjustment to the weight between the labeled real data and the generated data, aiming to optimize the adversarial risk. Numerical studies are conducted to verify the theories and show the effectiveness of the proposed algorithm.

**Questions:**

1. Do the theoretical results still hold for $p=\infty$?

2.  Could the theoretical results generalize to multi-class classification?

**Limitations:**

The limitations and potential negative societal impact of the paper are properly addressed.

**Strengths And Weaknesses:**

I think this paper has the following strengths:

1. The theoretical analysis is novel and the results seem sound, though I don't check the proof of the theories carefully. It is important to understand why and how using unlabeled data can benefit adversarial training. It is also important to understand how the quality of the unlabeled data generator affects the adversarial robustness.

2. It proposes an algorithm that dynamically adapts the weight during the training of neural networks and shows its promising performance empirically.

3. It is well-written and the ideas are clearly presented. The related works are properly discussed.

However, this paper has the following weaknesses:

1. There are some mismatches between the theory and the actual experiments. For example, it only considers $p=2$ in the theorems while in the experiments, it sets $p=\infty$. Do the theoretical results still hold for $p=\infty$?

2. It only considers binary classification for the theoretical analysis. Could the results generalize to multi-class classification?

3. It puts simulations and most real experiments in Appendix. I think it should at least summarize the results and findings in the main body of the paper.

---

> ### Author Response · Authors · 2022-08-01
> **Response**
>
> Thank you for your constructive comments for our paper! We have some common response to all reviewers about our updates in the paper, and below are some answers for your specific questions:
>
> 1. Weakness, theory when $p=\infty$: Thanks for pointing out this! Our general results of Theorem 1 and the intuitions for the label cost (Section 4.2) and generator cost (Section 4.3) all apply to $L_\infty$ attack. In our revision, in the appendix, we provide simulation experiments in $L_{\infty}$ attack. All the empirical behavior observed in $L_{\infty}$ attack are the same as in $L_2$ attack.
>
> 2. Weakness, generalize to multi-class classification: As mentioned in our reply to Reviewer RigD (Weakness, scenarios where assumptions do not hold), the actual assumptions required by Theorem 1 are not as tight as the assumptions that appear in the paper. Therefore, Theorem 1 may hold for multi-class classification as long as the loss function has a good shape and the gradient and Hessian matrix are well-behaved. Besides, our intuitions in the label cost and generator cost are also applicable to multi-class classifications.
>
> 3. Weakness, experiments are in the appendix: We appreciate your suggestion. Because of the nine-page limit, we did not have enough space to put more experiment results in the main content. As mentioned in the common response, we added one extra page in the appendix (to be integrated into the main content in the camera-ready version). On this extra page, we added some experiment result summaries.

---

### Official Review · Reviewer_RigD · 2022-07-10

**Rating:** 6
**Confidence:** 3
**Soundness:** 3 good
**Presentation:** 3 good
**Contribution:** 3 good

**Summary:**

This paper investigated the phenomena when using simulated data to improve adversarial robustness and provided the theoretical analysis. Specifically, the author decomposed the adversarial risk used in adversarial training to explain why and how unlabeled data can help and how its quality affects the resulting robustness.

**Questions:**

1. Why perform Taylor expansion for Eq. 4?
2. When doing simulation, how do authors make sure they are aligned with your assumptions? For example, the datasets used are in sub-Gaussian distribution.
3. When do the assumptions A1 and A2 hold?
4. In what situation, the assumption would not hold. In this case, how would the robustness performance be affected?


**Limitations:**

Please refer to my review.

**Strengths And Weaknesses:**

Strengths:
The authors clearly shape the research questions and provide the theorem and experiments to verify their points of view. The theoretical analysis for to answer the interesting questions that artificially data helps robustness is the main contribution of this paper. This paper is also generally well-written, delivering the main message clearly.

Weaknesses:
The proofs for the proposed theorems/lemmas/propositions are all in the appendix, I would suggest including more details, especially the motivation and reasons when giving the theorem.

It is intuitive to ask if there have some scenarios that the theorems cannot explain; the authors use too many assumptions in giving the derivations, however, if the assumptions should be reasonable not for convenience. The discussion on counterexamples where the theorems might not hold would shed light on the paper.

Secondly, in my opinion, most of the equations are hard to understand and follow, this might be due to the loss of the explanation before giving the statement.

Third, the experiments seem to lose some baselines for comparison, it would be great if compare with more others.

Last, the paper requires careful proofreading, e.g., "minimiax" in line 185, “… is the current model” in line 183, etc.

---

> ### Author Response · Authors · 2022-08-01
> **Response**
>
> We appreciate your effort in reviewing our paper! We have some updates and common response to all reviews at the top. Below is a list to answer your questions:
>
> 1. Weakness, proofs in the appendix: Thank you very much for this useful suggestion! We did not provide this in the main content because of the page limit. As mentioned in the common reply, we added a proof sketch on the extra page in the full paper (in the supplementary material) and will move it to the main content in the camera-ready version.
>
> 2. Weakness, scenarios where assumptions do not hold: Conceptually, we expect to derive important theoretical insights under relatively simple model and these insights generalizes well under complex model (via empirical justifications). On the other hand, although we need strong assumptions regarding the shape of the loss function, moments of the loss gradient, and the moments and eigenvalues of the Hessian matrix, they can be relaxed to certain degree. To avoid explaining mathematical conditions too much and losing the focus on our main insights, we choose to keep our assumptions in our representation. Instead, we added a small section (Section D) in the appendix to explain this. We also explained the possible outcomes when the assumptions do not hold.
>
> 3. Weakness, equations are hard to follow: We appreciate you sharing your feelings about reading our paper. We would try to improve the readability of the formulas.
>
> 4. Weakness, lack of baselines: Our paper focuses on theoretical investigation of some phenomenons in adversarial training, so it is not a methodology development paper that aims to beat SOTA results. Therefore, most of our simulations are used to justify theoretical finding, rather than competing performance with other methods. There is one exception where we propose algorithm 1 to determine proper weights.
>
>     For the real-data experiments of this algorithm, based on RobustBench (https://robustbench.github.io/), the best robust accuracy of CIFAR-10 under $\mathcal{L}_{\infty}$ attack for WideResNet28-10 and WideResNet34-10 are around 62\% to 63\%, which is similar to the result in our paper (Table 3 in the appendix).
>
> 5. Q1, Taylor expansion in (4): The Taylor expansion in $R(\widetilde{\theta}(\epsilon),\epsilon)-R(\theta_\epsilon,\epsilon)$ aims to transform the difference in the loss into the distance between $\widetilde{\theta}(\epsilon)$ and $\theta_\epsilon$. This is a standard step to linearize complicated representations.
>
> 6. Q2, simulations v.s. assumptions: In our simulation studies, we use $X\sim N(\textbf{0},\Sigma)$ and $Y$ is designed based on Assumption A1. Therefore, the simulation scenario satisfies the assumptions.
>
> 7. Q4, when assumptions do not hold: As mentioned before, we added discussions on the some possible outcomes if the assumptions do not hold.
>
>     In addition, some of our numerical experiments consider scenarios where the assumptions do not hold. For example, in the right panel of Figure 4, the poor generator (correlated) is a scenario that the generator is related to the labeled data set $S_1$ but not good enough (i.e., also violates Example 2). In this case, there is a correlation between the generated data $S_2$ and $S_1$, and the green curve in Figure 4 is quite different from the red curve (a poor but independent generator).

---

> > ### Comment · Reviewer_RigD · 2022-08-09
> > **Post-rebuttal response**
> >
> > I thank the authors for answering the questions. One of my main concerns is that the motivation and reasons for giving the theorem are missing. However, I appreciate that the authors have adequately addressed most of my other questions. Therefore, I increase my score to 6.

---

> > > ### Author Response · Authors · 2022-08-09
> > > **Thank you very much!**
> > >
> > > We would like to thank you again for providing us with such a constructive and encouraging review! We will try to polish our paper to fully emphasize the motivation and make the mathematical formulas easier to understand in the camera-ready version.

---

### Official Review · Reviewer_Htxg · 2022-07-11

**Rating:** 5
**Confidence:** 3
**Soundness:** 3 good
**Presentation:** 2 fair
**Contribution:** 3 good

**Summary:**

This paper includes several theoretical analyses about introducing additional artificial data in adversarial training: its benefits and the relationship between the generator performance and training performance. The analysis starts from their main theorem that decomposes the access risk into two parts: label cost (from mislabeling the artificial data) and generator cost (from the poor performance of the generative model).

Then, the authors use this decomposition to investigate their two research questions further. The answer to the first question is that, assuming an ideal generator (with no generator cost), the excess risk after introducing artificial data is smaller than the excess risk of vanilla adversarial training.  This result shows the benefits of introducing artificial data in adversarial training. Also, the author decomposed the generator cost further to bias and variance terms and showed that the bias term is upper bounded by the dissimilarity between the data distribution and the distribution of generated data. Because the variance term converges to 0 as the number of generated samples grows, the result shows that we can reduce the generator cost by having a better-quality generative model. This paper's last contribution is the strategy of weighting the usual training samples and the introduced artificial samples.


**Questions:**

1. Where are the missing proofs for Proposition 2 and Proposition 3? If they do not need proof, please explain. I consider those proofs missing quite seriously, so if the proofs are in the paper but I missed them somehow, please let me know so that I can adjust the rating.
2. I recommend the authors proofread the writings once again. Some typos and grammatical errors that I spotted are as follows.

  - Line 83: “$\theta_\epsilon = \min R(\theta, \epsilon)$” -> “$\theta_\epsilon = \arg\min_\theta R(\theta, \epsilon)$” ($\min R(\theta, \epsilon)$ is the value of minimum risk, but $\theta_\epsilon$ must be the model parameter minimizing the risk.)

  - Line 175: “The simulations results in” -> “The simulations result in” (Grammar)

  - Line 480: “Figure B.1” -> “Figure B.2” (It looks like that Figure B.1 is the result for Section B.2)

  - Line 516: “Lemma 1” -> “Theorem 1 (According to the structure of this section, the proof for Theorem 1 comes last.)

  - Equation after Line 547: “$\mathcal P_a \otimes \mathcal P_\epsilon$” -> “$\mathcal P_a \otimes \mathcal P_y$” (I don’t think that $\mathcal P_\epsilon$ is defined in the article.)

3. I don’t understand why the authors separated Section 5 from subsection 4.4, whereas it can be just a continuation of subsection 4.4. Also, it would be better to separate subsection 4.1 and the other parts, because subsection 4.1 looks to be the main insight and the other parts are analysis/design from the main insight.
4. In my opinion, you use the notation "$\cdot\otimes\mathcal P_y$" multiple times, consuming too much space in the paper. Defining a shorter notation for "$\cdot\otimes\mathcal P_y$" and changing all the occurrences would save some space.


**Limitations:**

This paper does not have a part assigned to address the limitations and potential negative societal impact. I understand the hardship of putting many results in the page limit, but it must be possible to condense the contents further to ensure space for this. (I don’t know whether moving some contents to the Appendix at this stage is allowed, but I recommend it if it is allowed.)

**Strengths And Weaknesses:**

Originality: To the best of my knowledge, the paper contains novel ideas.

Quality:

[[Strength]]
1. Assuming the correctness of lemmas, the proof of Theorem 1 seems correct.

[[Weakness]]
1. The proofs for Proposition 2 and Proposition 3 are missing. I don’t think they are trivial statements, but the proofs are neither in the main part nor the Appendix.

Clarity: There are a few typos and grammatical errors. See Questions for more details.

Significance:

[[Strength]]
1. This paper is dense with theoretical discussions on adversarial training. Considering the lack of theoretical understanding of adversarial training in adversarial machine learning research, I believe that this paper provides valuable insights into the field.

---

> ### Author Response · Authors · 2022-08-01
> **Response**
>
> Thank you very much for reviewing our paper! Besides the common response to all reviewers, below is a list of answers for your specific questions:
>
> 1. missing proofs: We did not provide the proof of Proposition 2 and 3 in our submission. For Proposition 2, it is only a simple extension of Theorem 1. In terms of Proposition 3, its statement already implies how we prove it. The last sentence, "a larger $n_2$ always gives a better $\widetilde{\theta}$", is the key conclusion for Proposition 3, and the other sentences are logic derivations leading to this conclusion. We added a short proof for Proposition 2 in the appendix Section F.3.
>
> 2. typos: Thank you very much for figuring out our typos, and we have corrected them in the main content (the full version in the supplementary material).
>
> 3. paper structure: We appreciate your suggestion on our paper structure. Due to the current nine-page limit, we cannot make big changes to the main content now. We will consider this in the camera-ready version.
>
> 4. Q4, notation $\otimes$: Thank you for pointing out this issue! We will take this into account in the camera-ready version.
>
> 5. Limitations, negative social impacts: Our paper mainly focuses on the theories and uses public data sets in our real-data experiments. We are not aware of any direct negative social impact by our results. We mentioned this in the checklist on Page 12.

---

> > ### Comment · Reviewer_Htxg · 2022-08-06
> > **Response to the authors**
> >
> > 1. missing proofs: I can see how the proof for the Proposition 3 works. However, however simple the proof is, I still believe that you should have a proof in the appendix or at least a short line about why the risk becomes smaller.
> > 2. Thanks for correcting the typos.
> > 3. I can understand that it is hard to make a big change.
> > 4. I can understand that it is hard to make a big change.
> > 5. I think that you are encouraged to create a separate "Limitations" section "in your paper" (see https://neurips.cc/public/guides/PaperChecklist) and Page 12 is not a part of the main body in your paper.

---

> > > ### Author Response · Authors · 2022-08-07
> > > **Thanks for your response**
> > >
> > > Thank you so much for your response! We updated the full paper (in supplementary material) with the new section "Limitations of This Work" (new Section A in the appendix) and the proof for Proposition 3 in Section G.3 in the appendix. As mentioned in the common response before, we will modify the main content to add essential connections to the new things in the appendix in the camera-ready version.
> > >
> > > Please let us know if you have further suggestions or questions.

---

### Author Response · Authors · 2022-08-01
**Common Response to All Reviewers**

We greatly appreciate the reviewers reviewing our paper and providing many insightful suggestions.

We update the full paper (with the appendix) in the supplementary material to fix the minor issues and address reviewers' concerns. Below is a summary of important updates:

1. We update a new Section A.2 to do simulation studies under $\mathcal{L}_{\infty}$ attack as a part to address the weakness mentioned by Reviewer bXme besides the theory part. Briefly speaking, all the observations are the same as for $\mathcal{L}_2$ attack.

2. A new Appendix Section D is updated to explain how Assumption A1 can be relaxed and the possible outcomes when the assumptions do not hold. This new section aims to answer the questions of Reviewer RigD and Reviewer bXme. In short, when the loss has a good shape and the data distribution is well-behaved, our results can be applied to other scenarios, e.g., other loss function, or multi-class classification.

3. A short proof of Proposition 2 is added in Section F.3 to address the concern of Reviewer Htxg.

4. We add in Appendix Section E some new figures similar to Figure 2 but with changing $n_2$ based on the comment of Reviewer VSRZ. Briefly speaking, with $n_2/n_1$ increases, for an ideal data generator, the label cost gets larger and the generator cost is reduced. For poor data generators, it is hard to effectively reduce the generator cost.

5. Due to the nine-page limit in the revision stage, we add one extra page of content at the beginning of the appendix to include the proof sketch (Reviewer RigD) and some summary of numerical experiments (Reviewer bXme). We will integrate this page to the main text in the camera-ready version.

    * To summarize the numerical results, we conduct various simulations and real-data experiments to verify the correctness of our theory.

    * For simulation, we verify: (1) given the ideal data generator, the performance of $\widetilde\theta(\epsilon)$ is better than $\widehat{\theta}(\epsilon)$ when $\epsilon$ deviates from zero; (2) the better quality of the data generator implies the better performance of  $\widetilde\theta(\epsilon)$; and (3) balancing the weight between $S_1$ and $S_2$ improves the performance.

    * For real-data experiments, we verify that the label cost and the generator cost are important factors in deep learning. We show (1) adding more unlabeled samples from the ideal generator will improve adversarial robustness, and (2) adding unlabeled samples from a poor generator with a small $n_2$ will slightly improve the performance.

Currently, these major changes are not reflected in the main text due to the strict page limit. In the camera-ready version, we will update the main text accordingly, e.g., add related discussion or remarks.  All the revision changes in the supplementary material are highlighted in blue color.

---

### Meta-Review · Area_Chair_52Bi · 2022-08-23

**Recommendation:** Accept
**Confidence:** Certain

**Metareview:**

The recommendation is based on the reviewers' comments, the area chair's personal evaluation, and the post-rebuttal discussion.

This paper studies how synthetic data can be useful for improving adversarial robustness. All reviewers find the results convincing and valuable. The authors' rebuttal has successfully addressed the reviewers' concerns. Given the unilateral agreement, I am recommending acceptance

**Award:**

No

---

### Decision · Program_Chairs · 2022-09-14

Accept